# When performance is not enough—A multidisciplinary view on clinical decision support

Roland Roller[1,2]*, Aljoscha Burchardt[1], David Samhammer[3], Simon Ronicke[2], Wiebke Duettmann[2,5], Sven Schmeier[1], Sebastian Möller[1,4], Peter Dabrock[3], Klemens Budde[2], Manuel Mayrdorfer[2,6‡], Bilgin Osmanodja[2‡]

**1** German Research Center for Artificial Intelligence (DFKI), Berlin, Germany, **2** Department of Nephrology and Medical Intensive Care, Charité - Universitätsmedizin Berlin, Corporate Member of Freie Universität Berlin, Berlin Institute of Health, Humboldt-Universität zu Berlin, Berlin, Germany, **3** Institute for Systematic Theology II (Ethics), Friedrich-Alexander University Erlangen-Nürnberg (FAU), Erlangen, Germany, **4** Quality and Usability Lab, Technische Universität Berlin, Berlin, Germany, **5** Berlin Institute of Health, Berlin, Germany, **6** Division of Nephrology and Dialysis, Department of Internal Medicine III, Medical University of Vienna, Vienna, Austria

‡ MM and BO contributed equally to this work (shared last authorship).
* roland.roller@dfki.de

**Data Availability Statement:** The data which has been used for the study cannot be shared publicly in its current form as they include identifying patient information. The distribution is prohibited

## Abstract

Scientific publications about the application of machine learning models in healthcare often focus on improving performance metrics. However, beyond often short-lived improvements, many additional aspects need to be taken into consideration to make sustainable progress. What does it take to implement a clinical decision support system, what makes it usable for the domain experts, and what brings it eventually into practical usage? So far, there has been little research to answer these questions. This work presents a multidisciplinary view of machine learning in medical decision support systems and covers information technology, medical, as well as ethical aspects. The target audience is computer scientists, who plan to do research in a clinical context. The paper starts from a relatively straightforward risk prediction system in the subspecialty nephrology that was evaluated on historic patient data both intrinsically and based on a reader study with medical doctors. Although the results were quite promising, the focus of this article is not on the model itself or potential performance improvements. Instead, we want to let other researchers participate in the lessons we have learned and the insights we have gained when implementing and evaluating our system in a clinical setting within a highly interdisciplinary pilot project in the cooperation of computer scientists, medical doctors, ethicists, and legal experts.

## Introduction

The United Nations' Sustainable Development Goal 3 "Good Health and Well-being" aims to ensure healthy lives and promote well-being at all ages, e.g., by improving access to physicians and information. Artificial intelligence (AI)-based tools, such as decision support systems, are

by the data protection officer at Charité - Universitätsmedizin Berlin. The data contains not only text data but also much detailed information about single patients over time, including dates (when the patient came around for a visit or was hospitalized), his/her diseases, the treatments, etc. In addition, some phenomena in the data might be particular, and therefore it might be easier to re-identify a single person. Requests for data access can be sent to the Data Protection Officer at Charité Universitaetsmedizin Berlin: Janet Fahron, email: datenschutzbeauftragte@charite.de. We plan to de-identify multiple aspects of the data so that it can be shared, given a reasonable request from other researchers. The model, instead, can be shared on request.

**Funding:** The project and all authors received funding by the German Federal Ministry of Education and Research (BMBF) through the projects PRIMA-AI (01GP2202C) and vALID (01GP1903C) https://www.gesundheitsforschung-bmbf.de/de/valid-klinische-entscheidungsfindung-durch-kunstliche-intelligenz-ethische-rechtliche-und-10430.php And no, the sponsor did not play any role in the study design.

**Competing interests:** The authors declare that there are no competing interests.

expected to contribute to this goal, as they aim to increase diagnostic precision and improve the monitoring of medical treatments. Even though the healthcare sector comes with many well-justified constraints regarding the use of data or the design of experiments, it can be considered one of the drivers of applied research in AI and machine learning (ML) in particular. In the past, many scientific publications in ML evolved around the introduction of novel methods and boosting performance, e.g., of classification or prediction tasks based on readily available data [1–3].

While implementing methods for the best possible system performance is important and necessary, it is only the first step and may sometimes be seen as a low-hanging fruit. Many fundamental research questions subsequently emerge when the goal is to make practical use of such a system in healthcare: Can it be further developed without medical expertise and if not, how and where does it come into play? How can an evolving system be evaluated? What are the requirements for the assessment of transparency and trust? How does the system affect the relationship between medical experts and patients? Given the complexity and the abundance of aspects to be taken into account from different domains, it seems obvious that these questions cannot be answered by computer scientists alone and likewise they cannot be answered by medical or ethical experts on their own without considering technical expertise.

To move a step forward, this article is written for computer scientists working (and particularly starting to work) at the intersection of AI and medicine. First, we describe the development and validation of a risk prediction model designed to assist physicians by predicting the risk of severe infection in kidney transplant recipients (KTR). This is important since KTRs are at high risk of infection due to immunosuppressive treatment with concomitant diagnostics. Balancing the risk of infection and the necessary immunosuppression to prevent rejection of the kidney transplant is challenging even for experienced transplant physicians. This is why such a risk prediction model can potentially support physicians [4].

Based hereon, we discuss the main challenges and lessons learned from a technical perspective. In doing so, we take medical and ethical perspectives into account and try to shed light on aspects that are worth taking into consideration to get beyond a simple performance gain. Most related work reporting about challenges and learnings in this domain discusses the topic from a less practical perspective, drawing comparisons to a large range of existing publications rather than drawing conclusions from their development [5–7]. The main contributions of this paper are:

- An internal evaluation of a straightforward risk prediction model for severe infection in kidney transplant recipients that is implemented in a clinical patient database.

- A task-based evaluation of the system in a reader study with physicians at different levels of experience.

- A description of specific lessons learned during the implementation and evaluation of the system in the clinical setting.

- A description of general insights gained in the research within the multidisciplinary cooperation of computer scientists, medical doctors, ethicists, and legal experts.

## Related work

During the last few years, research output about AI-based decision support systems in medicine has increased considerably [8]. The challenges that developers, physicians, and regulators have to overcome depend, among others, on the designated users (e.g. pharmaceutical

industry, physicians, patients) and the targeted medical domain. In the area of kidney transplantation, which we are studying in this article, physicians use a combination of demographic, clinical (including microbiological and medical imaging), and mostly laboratory data to implicitly risk stratify patients according to their risk for rejection, infection, and graft loss and to adapt the immunosuppressive treatment accordingly [9].

Most existing decision support systems or prediction models in the context of kidney transplantation focus on predicting either kidney function (creatinine-based estimated Glomerular Filtration Rate (eGFR)) or graft loss and incorporate mostly structured data [10–13]. This differs considerably from other domains, where image recognition plays an important role (radiology, pathology). Most work which is related to the prediction of infectious complications in kidney transplantation does not use machine learning methods, but classical statistical methods [14–18]. A summary of the relevant studies in the field of kidney transplantation is provided in Table 1.

A general problem is that most research stops after model development and validation. Only for the iBox model, a reader study was performed, comparing the prediction algorithm to physicians [20]. They show on the one hand that physicians tend to overestimate the risk of graft loss and are rather inconsistent in their predictions, and on the other hand that the prediction model outperforms the physicians. However, the performance implications of prediction algorithm-assisted decisions by physicians remain open. This is a blank space in research on AI-based decision support in general, not only in kidney transplantation. Therefore, we chose not only to develop and validate an AI-based prediction model for severe infection in kidney transplant recipients. We also aimed to analyze how the performance of physicians changes after incorporating the system's predictions into their decision-making. We argue that this step is of greatest relevance since the sociotechnical environment into which an AI system is embedded will influence performance in clinical practice.

In other medical domains, human-machine interaction has been given more attention so far. The work of Sutton et al. [21] provides a good introduction to clinical decision support and discusses important challenges, such as the implications of too many false positives. Zicari et al. [22] discuss the development of an ethically aligned co-design to ensure trustworthiness. However, the work remains more theoretical, with less specific practical advice as the work

**Table 1. Risk prediction models in the domain of kidney transplantation.** Perf.—Performance during external validation or optimism corrected for internal validation, Ref—Reference, ANN—artificial neural network, RNN—recurrent neural network, LASSO—least absolute shrinkage and selection operator, LR—logistic regression, Cox PH—Cox proportional hazards regression, RF—random forest, GBRT—gradient boosted regression trees, Tx—transplantation, D development dataset, V: external validation dataset, D/V: development and internal validation.

| Prediction Task | Patient count | Method | Perf. | Ref. |
|---|---|---|---|---|
| eGFR at upcoming visit | D/V: 675 | ANN | MSE: 99 mL/min/1.73 m2 | [13] |
| eGFR within 3 months after Tx | D: 933 V: 1170 | RNN (sequence-to-sequence model) | root MSE: 6.4-8.9 mL/min/1.73 m2 | [12] |
| Graft Survival | D: 3774 V: 9834 | Bayesian, shred-parameter, multivariable joint models | AUC 0.820–0.868 | [11] |
| Graft Survival | D/V: 3117 | Survival decision tree | C-index 0.71 | [10] |
| Infection (general) within 3 years after Tx | D/V: 863 | LASSO LR based nomogram | AUC 0.81 | [15] |
| Infection (general) within 6 months after Tx | D: 410 V: 522 | LR | AUC 0.73 | [14] |
| Infection + rejection in the first year after Tx | D: 1190 V: 630 | Cox PH | n/a | [16] |
| Pneumonia in the perioperative period | D/V: 519 | RF | AUC 0.91 | [18] |
| BK-virus infection from 2.5-8.5 months after transplantation | D/V: 312 | dynamic Cox PH | AUC 0.70 | [17] |
| COVID-19 vaccine response | D: 590 V: 952 | LASSO LR, GBRT | AUC 0.812 | [19] |

reports on an early developmental stage and thus misses the aspect of a real system and experiments to discuss. In Amann et al. [23], the authors came to a similar conclusion as we do that performance alone is not enough, and therefore stress the importance of explainability. The authors discuss this topic from multiple perspectives but provide only an overview of the topic without getting too much into detail. Bruckert et al. [24], again a high-level and interdisciplinary work, present a roadmap to develop comprehensive, transparent, and trustful systems, considering the "black-box nature" of many machine learning approaches. However, the work mainly targets explainability and interaction and does not validate the ideas within a particular experiment.

Although not too recent, the work of Topol [8] provides an overview of different medical fields and the most relevant work regarding the developments of AI algorithms. The authors also present publications from different medical fields in which the performance of AI methods has been compared to the performance of doctors. Moreover, the author highlights that AI can enhance the patient-doctor relationship—a perspective that has not been analyzed in detail in our work. Finally, the work of Ho et al. [25] arrives at a similar conclusion to ours, that the inclusion of multiple stakeholders is necessary to develop AI technology in healthcare.

Concluding, there is a large range of publications addressing the aspect of automatic AI-driven clinical decision support (or particular aspects of it), its challenges, and coming to partially similar conclusions as we do, such as explainability vs. interpretability [26], trust [6, 27], data, or evaluation issues [5, 28]. However, this work discusses multiple aspects of developing such a system from a rather technical perspective while taking into account also additional aspects which are necessary for realistic implementation. Moreover, the discussion does not remain theoretical, as we substantiate it with challenges and learnings we have experienced along a use case in nephrology.

## Data and methods

The risk prediction model we present here was developed in close collaboration between computer scientists and clinical experts. The main goal of our risk prediction model was the detection of kidney transplant patients at risk of an infection within the next 90 days expressed as a risk score between 0 and 100. Kidney transplantation requires immunosuppressive medication, and while too little immunosuppressive medication can lead to rejection and transplant failure, the majority of patients suffer from the opposite—infections. Severe bacterial infections are strongly associated with a very high CRP (C-reactive protein) level in the blood, e.g., a CRP level increases from a normal value below 5 to above 100 mg/L.

### Data and task

The data used for our work is an electronic medical record (EMR) called TBase [29], which has been developing for 20 years. TBase has more than 4500 long-term data sets of KTR including laboratory values, vital parameters, medication, diagnosis, operational information, clinical notes, medical reports, etc. The data are partly structured and partly unstructured. As transplanted patients need to come to the hospital for checkups 3-4 times a year, Tbase contains a lot of valuable information on a fine-grained level over many years.

Patient data from 2009-2019 was included. Data itself was selected according to data points, which we define as a point in the life of a patient documented in the database. Each time a new entry is made in the patient database for that patient, a new data point is created. A datapoint defining an infection within a time window of 90 days was labeled as "true", and otherwise as "false". The same has been done for 180 and 360 days. Next, some filtering steps have been applied in which data points of patients who are currently experiencing an infection, as well as

patients of less than 18 years of age, were filtered out. Furthermore, the first two weeks after kidney transplantation are dismissed, as operations that have an impact on inflammatory lab values within that period. Finally, only data points with a follow-up data point within the next 15 to 180 days are used. This filter has been implemented to exclude gaps in patient follow-up ensuring the reliability of endpoint evaluation.

Given a data point, including previous information until that point in time, the risk prediction model estimates how likely an infection occurs within a given time frame. Note, the data is unbalanced—about 12% of the data points include infection within the next 90 days.

**Infection** was defined based on CRP levels in the blood. Since CRP elevations can be caused by other non-infectious causes (e.g. rheumatic diseases or cancer) and to focus on relatively severe infections a threshold of 100 mg/L has been set for endpoint definition.

## Methods

For our implementation, we use a Gradient Boosted Regression Tree (GBRT), implemented in python using scikit-learn, which can be trained quickly, also on less performant infrastructure. Overall, our model was intended to be a baseline for a first prototype. It was developed within the hospital, given the available infrastructure, in close collaboration with medical experts. Before training, the data was preprocessed, transformed, and filtered. This included, for instance, dealing with missing information (in our case the model worked better if missing data was not automatically inserted) and processing noisy fields in the database (e.g. converting different units for lab values or numeric fields which contain text data). Moreover, as machine learning models often struggle with unbalanced data (ratio between positive and negative examples), we randomly downsampled the majority (negative) class in the training data to a ratio of 1:3 (positives:negatives), which resulted in the best system performance during initial experiments for our setup. Controlled upsampling or downsampling strategies, such as SMOTE [30] or NCA [31] did not result in significant improvements in comparison to the approach. An overview of the generation of the risk prediction system—from data to model—is presented in Fig 1.

The final GBRT model relies on 300 estimators with a maximum depth of 3, and a learning rate of 0.1. The model integrates about 300 different features, consisting of lab values, vital parameters, diagnoses, medications, demographics, etc. The model was trained on a Ubuntu 18.04.03 LTS server with an Intel Core i9-7900X 3.30GHz, which was located inside the hospital. The resulting model, as well as further descriptions of the data and usage of the model, can be made available on request. Note, the model itself has already been tested in a different setup, to detect patients at risk of transplant loss or rejection [32].

The studies involving human participants were reviewed and approved by the Ethics Committee of Charité—Universitätsmedizin Berlin (EA4/156/20). Written informed consent for participation was obtained for all participants in the reader study.

## Results

Our model predicting patients at risk for future infections has been evaluated in two different setups: (1) on retrospective data, firstly within an internal validation using cross-validation, and (2) within a reader study—an evaluation of the model together and "versus" physicians. Both experiments are described in the following chapters.

### Internal validation

Within an internal validation, we evaluated our system using a 10-fold cross-validation on the historic, retrospective patient data. Each time, the data were randomly split into training,

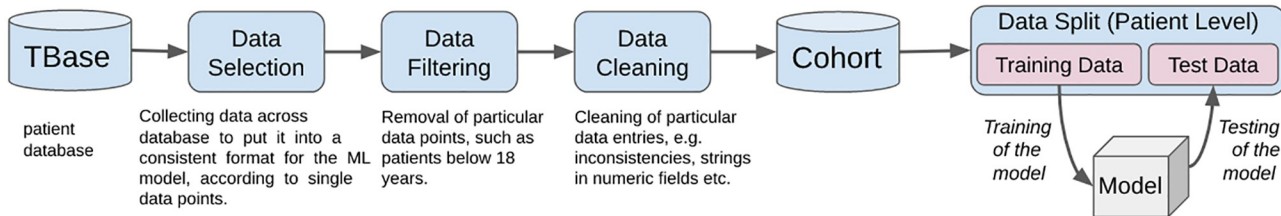

**Fig 1. Overview of the data flow from the patient database to the model.** The last part about the patient-level data split depends on the evaluation. In the case of internal evaluation, the data split represents the 10-fold cross-validation. In the case of the reader study, the split has been applied just once.

development, and test using a split of 70/15/15. The split was conducted on a patient level, to ensure that no data point of the same patient will occur across different splits. In addition to the endpoint prediction for the next 90 days, we also explore the prediction within the next 180 and 360 days. Even though Area under Receiver Operator Curve (ROC) is frequently used for the evaluation of this kind of task [8], it is known that ROC cannot deal very well with unbalanced data [33], which is often the case in a clinical setup. For our evaluation, we use Receiver Operator Curve (ROC), as well as Area under Precision-Recall Curve (PRC).

The results of the internal validation are presented in Table 2 and show how well the model can forecast an infection within the next 90, 180, and 360 days. For the intended prediction within 90 days, the ROC score is 0.80 and the PRC score is 0.41. In comparison, with the longer time frames, the ROC score is a bit higher for the event in the near future while the PRC score increases for the long-term predictions. As there is not much work, targeting similar problems and at the same time, using similar data, a comparison is difficult.

## Reader study

Based on the internal validation, it was difficult to assess the quality and efficiency of the system as to the best of our knowledge no related work addressing this problem existed. Therefore it was also not clear if this model would be useful for medical experts in some way or another. Thus, instead of focusing on the further optimization of our machine learning model, we decided to find out how well physicians can solve the given task, and if our approach might benefit them already.

To this end, we designed and conducted a reader study, consisting of two parts: In the first part, physicians received a data point of a patient, together with all information up to this time, and had to make a risk estimation (0 to 100%) if an infection will occur within the next 90 days. In the second part, the physician additionally received information on the risk prediction model, in the form of a dashboard (see Fig 2). The dashboard depicts the risk score over time within a graph, mapped to a traffic light system, indicating if the patient has got a low, medium, or high risk of infection, along with local and global features of the risk prediction model. Whereas global features are generally important features for the overall model, local

**Table 2. Result of 10-fold cross-validation of the automatic risk prediction system (GBRT) to predict the endpoint "infection" on retrospective data, according to Area under Receiver Operator Curve (ROC) and Area under Precision-Recall Curve (PRC).**

| Days | ROC | PRC |
|---|---|---|
| 90 | **0.80** | 0.41 |
| 180 | 0.79 | 0.46 |
| 360 | 0.77 | **0.50** |

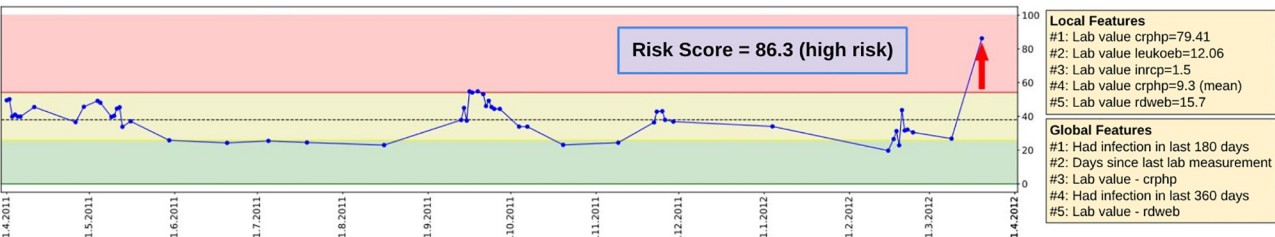

**Fig 2. Overview of the dashboard which shows the recent risk score (red arrow), together with the previous risk scores of the patient over time depicted in a graph.** Moreover, the dashboard shows the most important local features, which had a direct influence on the current scoring of the model, as well as the most relevant global features, which had generally a strong influence on the model.

features are those which had the strongest impact on the current prediction of the system. The cutoff of values for the traffic light system was calculated by using the predictions on a separate development set. While the dashed line in the middle represents the threshold for the optimal F1 score, the yellow area is defined by F2, which highlights recall over precision, and the red area is defined by F0.5, which highlights precision over recall.

While automatic risk estimations can be generated within seconds, humans require time to carry out this task. This fact indeed limited our possibilities for human evaluation. Overall, our study included 120 data points of 120 different patients (38 infections, 82 without infections). Eight physicians participated in our study, four experienced (senior) and four assistant (junior) physicians. Within each part of the study, each physician received 15 data points to analyze. The study was accepted by the ethical advisory board of the hospital, and according to the staff council each physician had up to 30 minutes to examine the data of each patient. Each physician received some basic introduction about the dashboard, information about how the machine learning model works, as well as its performance on the internal validation study. In each part of the study, each physician received different data points to evaluate as we wanted to prevent priming effects. This has to be taken into account when both conditions (without/ with AI) are compared. Finally, the results of both parts of the study are then compared to the performance of the risk prediction system itself. In the following, we refer to the first part of the study as MD (medical doctor) and the second part as MD+AI (medical doctor including machine learning support). The machine learning component itself will be referred to as AI. The experimental setup of the reader study is depicted in Fig 3.

For the experiment, we re-trained the model from the internal validation and made sure that no patient of the test set occurs within the training and development set. The evaluation of the experiment is carried out according to ROC and PRC, as well as sensitivity (recall), specificity (true negative rate), and positive predictive value (PPV, or precision) using different cutoff thresholds. The cutoff values were necessary to map from a regression score between 0-1 to a binary outcome (true/false), to calculate sensitivity, specificity, and PPV.

The results presented in Table 3 show that overall the task is challenging for physicians, considering the ROC score of 0.63. Moreover, the results indicate that physicians receiving additional automatic decision support do not increase performance. On the other hand, the risk prediction system outperforms the physicians in both parts of the study regarding ROC and PRC.

While the ROC and PRC results of AI show a promising improvement in comparison to the physicians, this does not tell us if patients would be labeled as critical (infection will occur) or uncritical. Thus, we use different cutoff values to map from the probability estimation of the physicians and the regression score of the risk prediction system, to either 1 (infection) or 0 (no infection). In this way, we can evaluate the risk prediction system, as well as the physicians

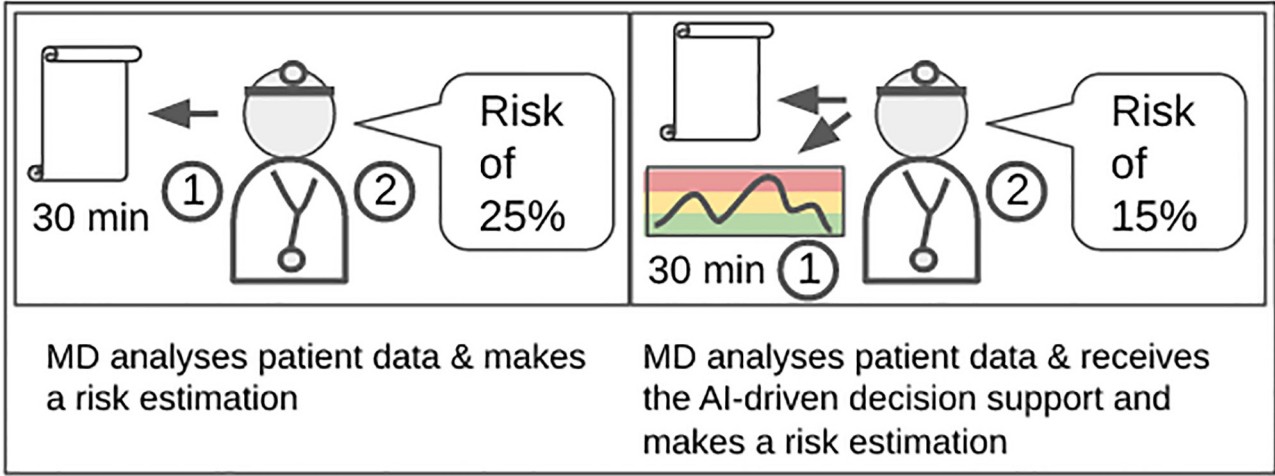

**Fig 3. Overview of the two parts of the reader study.** On the left, the setup with medical doctors (MD) alone. The MD has up to 30 minutes to 1) study the patient at a given date in his/her life and then 2) make a risk estimation for infection for the next 90 days. The right-hand side depicts the setup in which 1) the MD also receives the patient data and the risk estimation of the decision support system. After analyzing both for up to 30 minutes, the physician 2) makes a risk estimation.

**Table 3. Results of the reader study which shows the performance of physicians predicting infections within the next 90 days—without (MD) and with (MD+AI) decision support, in comparison to the automatic risk prediction (AI).** Evaluation is carried out using ROC (Area under Receiver Operator Curve) and PRC (Area under Precision-Recall Curve), as well as SEN (sensitivity; recall), SPEC (specificity; true negative rate), and PPV (positive predictive value; precision) defined by different thresholds (thrs). The threshold defines the cutoff value to map from regression to binary classification.

|  | ROC | PRC | SEN | SPEC | PPV | thrs |
|---|---|---|---|---|---|---|
| AI | 0.719 | 0.567 | 0.754 | 0.718 | 0.264 | F2 |
|  |  |  | 0.580 | 0.844 | 0.333 | F1 |
|  |  |  | 0.329 | 0.940 | 0.424 | F0.5 |
| MD | 0.630 | 0.474 | 0.448 | 0.689 | 0.406 | >50% |
| MD+AI | 0.622 | 0.469 | 0.368 | 0.829 | 0.500 | >50% |

according to sensitivity, specificity, and positive predictive value. The results in Table 3 show that AI achieves the highest sensitivity, therefore finds the largest number of critical patients, depending on the selection of the cutoff value. All results present only a moderate performance according to the positive predictive value (PPV). In the case of MD+AI for instance, every second prediction would be a true positive infection prediction, while the sensitivity is about 37%.

Table 4 compares the ROC performance of the two subject groups in both parts of the study. The table shows that senior MDs reach higher values than junior MDs at solving the task alone, without decision support. Moreover, the table indicates that junior MDs increase in performance, together with the automatic decision support, while senior MDs decrease.

**Table 4. AUC-ROC performance of the two different subject groups: Physicians without (MD) and physicians with (MD+AI) automatic decision support.** The subject groups are divided into junior (less experienced) and senior physicians.

|  | Junior MD | Senior MD |
|---|---|---|
| MD | 0.5781 | **0.6772** |
| MD+AI | **0.6398** | 0.6149 |

## Discussion

Our initial experiments have provided first and promising results regarding the risk estimation of infections in KTR. The following conclusions specifically target computer scientists who will be involved in future clinical decision support systems. In addition to that, we then discuss further implications and learnings, highlighted from a multidisciplinary view (technical, medical, and ethical).

### Learnings made during the development

**Development constraints.** In general, many circumstances affected the development of the risk prediction model. The development had to be conducted within the hospital infrastructure and network, due to the sensitivity of the data. This meant that limited computing power was available, partial admin rights of the working computer were missing, and access to the internet was restricted.

**Data constraints.** Real clinical data tend to be noisy. In our case, some older data did not have the same quality as more recent data points. Certain database fields were not used anymore, were used to enter other information as the field was coded for, or were used only occasionally. This means, domain expert knowledge was required to interpret and subsequently filter and normalize the data, including lab values that were recorded in different units and occasionally were measured with different techniques over time. Here a narrow collaboration with the physician is helpful.

**Never trust a single score.** A somewhat trivial piece of advice has proven helpful: Never trust a single score. While a ROC score of 0.80 in the internal validation appears to be promising, it is surprising that the 180 and 360-day predictions achieve similar results, as a prediction in those long-term time frames seems to be quite difficult. One reason could be that the model might be able to identify factors, which generally increase the risk for infections, rather than those predicting a particular problem for the near future. In addition, the prediction of an infection within the next 360 days, also includes events that occur already in the near future. Moreover, the PRC score for 90 days is only mediocre, showing an even stronger counterintuitive increase for longer time frames.

**The issue with false positives.** Depending on how the threshold is chosen, the results of sensitivity, specificity, and PPV change. From a patient perspective, the presented results are far away from optimal. None of the approaches will predict all possible patients at risk, and the more critical patients can be detected the lower the PPV. Can such a setup be possibly useful for clinical practice? Particularly if the number of false positives is too large the automatic prediction might not be seen as trustworthy anymore or might lead to "alert fatigue" [34]. The best methods are still lacking as reporting of non-perfect machine learning scores on medical prediction tasks appears to be common and no system provides a perfect score [35, 36]. However, reporting results without taking the medical expertise into account, and without knowing the benefit might be not optimal.

Another aspect that fits here is the fundamental question of whether a binary prediction (true/false) of an infection within the next 90 days is the best possible support for medical doctors. From a computer scientist's perspective, such predictions and evaluations based on it are straightforward, but doctors might have questions like: Does this patient need any treatment now? When do I need to see this patient next? That requires different types of reasoning. Also, the current binary evaluation cannot take into consideration if a patient would suffer an infection after 91 days. In this case, if such a patient would be identified as high risk (high score), it would be considered a false positive in the evaluation.

**Data analysis.** The usage of multiple scores is certainly beneficial, to an overall impression of the system quality. On the other hand, looking into the data, and analyzing the estimations of single patients is essential. Is there something the automatic risk prediction can do that the physician cannot do? Or can only the obvious cases be predicted?

In our experiment, for instance, a detailed analysis revealed that the risk prediction system can partially detect different patients at risk. More precisely, using a threshold of 0.5 for physicians and F1 for the machine learning model, 29 future infections, out of 38, can be detected—either by AI or/and MD. Whereas nine future infections are not detected at all. Notably, while there is an overlap of 16 infections that can be detected by both, AI and MD, there are two cases that are only detected by a physician and eleven which are only detected by the AI. Instead, MD-AI does not lead to any further novel cases which have been either detected by AI or MD themselves. This insight is a strong argument for using the system as a double-check for physicians.

**Communication of results.** Even though our model outperformed physicians on the risk prediction task, the experiment showed that physicians may not improve with assisting automatic predictions. This shows that performance alone is not sufficient to introduce novel technologies into clinical practice. Moreover, as we have seen in Table 4, junior MDs increased in ROC performance, while senior MDs did not. This indicates that the human-machine interaction and in particular the way how information is communicated might be essential and need to be further studied for the success of the system. Besides the optimization of the dashboard and the communication of system explanations, the overall integration of such a system might be crucial. We believe that automatic decision support might be valuable as a double check, particularly after regular treatment. In this way, a physician has already got an opinion and might not be (possibly negatively) influenced even before analyzing the patient data in the first place. After the treatment, an automatic alarm could be raised, but only in case the system has detected a risk, the physician has missed. Future research in this area is needed.

## More general learnings

### Communicate with relevant stakeholders

To develop machine learning models on clinical data, communication is key. Frequent interactions with relevant stakeholders certainly help to understand the task and to identify relevant factors to solve it or to understand which information is reliable (or has to be ignored) in the database. As simple as this sounds, it is not, as for instance physicians have got a different background and so do you. It is necessary to find a "common language". In an ongoing project not reported here, we additionally take the patient's perspective into account. While this is undoubtedly important, it comes with another bunch of challenges.

### Start with a simple system

From a machine learning perspective, the selection of the underlying model has a huge influence on the outcomes. Therefore, neural models seem to be the first choice. However, in most cases, we recommend starting with a simple system to develop a solid and reliable baseline first. A model which can be quickly developed, and quickly trained, also on less powerful computing architecture—as we did. Moreover, starting generally simple, with fewer features and slowly increasing the complexity can be beneficial, as (sequential) medical data can be complex.

## Make use of expert knowledge

Machine learning engineers often think that a model solely needs to be trained on large data to solve the problem by itself. Data-driven ML models might be able to learn new or at least different correlations and patterns as compared to physicians who can reason in terms of causation. In our setup, the number of lab values appears to be a good indication that something is going wrong—conversely, physicians would probably rather argue that some values are "borderline" or abnormal, therefore additional lab examinations might be needed, which has been also observed in other related work [37]. Although different, both observations indirectly refer to the same—more lab examinations are needed, as something does not seem to be okay. In any way, we recommend taking the opinion of physicians into account (ideally from the beginning) and examining if their suggestions about relevant information/parameters contribute to the system's performance. For instance, is the suggested information covered by the data going into the model, or can the information properly be represented by the model (e.g. fluctuations of particular lab values)?

## Use case/target definition

For every research project in medicine, endpoint definition at the very beginning is crucial. Endpoints need to be precise and clinically relevant. When using clinical routine data to train machine learning algorithms, endpoints need to be defined from preexisting data, which can be tricky. While endpoints in prospective clinical trials are explicitly assessed by performing diagnostic tests at a certain time point defined in a study protocol (e.g. kidney biopsy one year after study inclusion), this is not the case for retrospective trials using clinical routine data. Hence, it is not advisable to use endpoints, for which diagnostic tests are not regularly performed. In our example, to predict infections, we chose a laboratory-based definition, because, for the majority of data points in our dataset, this laboratory value is available and always will be determined in case of severe infection. On the contrary, it would be much more problematic to predict a rare event, which requires a specific diagnostic examination that is not routinely performed. Every prediction based on such data will be imprecise because the diagnosis is often missed or at least delayed by the physicians themselves. It is important to keep in mind that no AI tool can detect cases, it has not learned to detect. Even with that in mind, defining a complex endpoint such as infection using a single laboratory parameter can be too simplistic and again not optimal for the clinicians. Specifically, our definition ignores less severe, and non-bacterial infections, but includes certain rare, non-infectious reasons for increased CRP value. Therefore, to prevent the necessity of redefining endpoints later in the model development, it is again advisable to consult several clinicians about their opinion about specific endpoint definition, as the endpoint definition not only has to meet clinical usability but also has to be accepted by the physician.

## Lack of comparability

The evaluation of a method is essential. On the other hand, comparability and reproducibility are major problems for AI in healthcare, as datasets are often not available or heterogeneous. Each data set has its characteristics. Achieving good results on one dataset is not a guarantee that you also achieve similar results on a different one, addressing the same problem. One reason for this is if the underlying dataset (cohort) is selected slightly differently and healthcare workers do not enter information standardized.

## Setting of the study and choice of study subjects

In addition to performance evaluation, most medical problems involve at least one medical expert—and potentially an AI-based system—for resolving the task. Thus, the interplay between

medical experts and AI-based systems needs to be considered in the evaluation. Human-machine interaction is just one aspect of this interplay. Importantly, the set-up of the experiments in which humans and the system collaborate, the information that is given to the human participants about the AI-based system, the setting of the study, etc. are all factors that are likely to impact the results. Thus, a simple comparison between human and machine baseline does not help estimate how such a sociotechnical system would perform in practice. If the patient's view is to be taken into account, depending on the state of the patient, it can be advisable to also include nursing staff, relatives, or even third parties such as patient representative organizations.

## Model explanations

The "explanations" of our system are based on local and global features—a simple and pragmatic solution. In many cases, this is a particular lab value, the presence of a diagnosis, the intake of a medication, or the time since the last transplantation. In contrast, in previous work, we explored how physicians would justify their risk estimations [38]. Those justifications were short paragraphs consisting of a few sentences and partially contain similar information as the relevant features of the ML system. However, justifications contain, aside from pure facts, also descriptions of fluctuation and tendencies of certain parameters ("lab value is raising since the last few months"), negations/speculations (e.g. explicit absence of symptoms), additional knowledge ("lab value above the norm", "patient does not seem to be adherent") and interpretations ("patient is fit"). Overall, medical decisions do not only depend on medical knowledge/experience and parameters provided in a database, but also on conversions to the patient or nursing staff. This information is only partially contained in clinical databases, if then only in clinical text. Moreover, we experienced that human justifications strongly vary, not only depending on task and risk score but also depending on the individual physician. This indicates that a "one-size-fits-all" perfect system explanation will probably not exist, as each physician might have their preferences. As understanding the system recommendation is essential to build up trust, the topic of communicating and justifying system decisions is still an open research domain—although much work is currently conducted in the context of XAI (explainable AI) [39].

## A customizable user interface

Another relevant aspect we experienced in our experiment is the wish for the personalization of such an automatic decision support system, which might be grounded in the need for professional communication mentioned above. The current approach is static, it is optimized according to ROC, and presents results within a dashboard. However, at the end of our experiments, physicians expressed an interest to interact and modify the system according to their preferences. For instance by excluding parameters, or increasing the relevance of some aspects. We assume that an automatic and interactive decision support system might help to easier understand the technology behind it and understand the potential, as well as limitations of such a system. In this way, a more efficient support system could be created, according to the wishes and needs of the treating physician, e.g., targeting only high-precision predictions, or achieving a larger recall. This might help to build up more trust in this technology. Techniques that are discussed under the heading of XAI like counterfactual explanations or the display of similar cases from the training data may further improve the usefulness of the system here. However, in any case, an overloaded dashboard is less helpful and should be avoided.

## From model to usage

From a clinical perspective, many applications of AI represent improvements on a technological level, but approach problems that are either mastered by human physicians or can and

have been solved with classical statistical methods as well. For AI researchers working with medical data, the long-term goal should be to arrive at a system that can improve patient outcomes or reduce physicians' workload. Only then it will be considered beneficial by evidence-based medicine or by physicians in daily practice. Great performance alone will not be enough for that. We think that systems aiming above all for successful translation into clinical practice will have a higher impact than very performant prediction tools for a suboptimal use case. Unfortunately, this is usually more difficult to achieve concerning data collection, preparation, and outcome definition. Additionally, structural barriers to research funding limit the possibility to follow long-term projects.

Nevertheless, for a tool to be eventually implemented in a clinical setting, design questions should be addressed carefully in the beginning. Specifically, use cases should be defined before the development of a tool by performing user research involving physicians, nurses, or patients at the beginning. They should be selected based on existing weaknesses in clinical settings and consider the problems and limitations of clinical work and reasoning. By choosing such cases, the potential benefit can be maximized and is easier to demonstrate in controlled trials studying the system's impact. Randomized controlled trials are the soundest from a medical perspective. Finally, close interaction with the end user during implementation is crucial as they have to be trained, not only on practical aspects but also on the strengths and weaknesses of the model to make an informed treatment decision.

## Connecting different stakeholders

It has become standard practice for better AI decision support systems to claim that they are aligned with the concept of trustworthy AI. Numerous criteria have been developed for this purpose, for which various reviews are now available [40, 41]. If one also takes the frame of the High-Level Expert Group on Artificial Intelligence: Ethics guidelines as a reference, one can say: Any AI-System must be compatible with applicable laws, meet ethical standards, and not entail unforeseen side effects [42]. However, trust is not a concept that is valid once achieved but must be constantly maintained [43]. Once achieved it must constantly be reviewed and maintained. Therefore, it is crucial to how the path from principles to practice is designed. Formally, approaches of co-creation AI [22] are well suited for this challenge. But also terms of content need to be unfolded. For this purpose, we decided in our experiment to analyze and evaluate the attitudes of the physicians involved.

Of particular interest for AI-driven support systems in the clinic is the question of how interaction processes change through the introduction of such and how normative concepts like trust, responsibility, and transparency have to be rethought in a new and interrelated way [44]. To approach these questions in the context of our experiments, qualitative interviews were conducted with the physicians at the end of the study. We decided to use semi-structured expert interviews as the data collection method. The interviews were intended both to obtain an evaluation of the conducted case study and to find out what impact the introduction of AI-driven support systems has on the interviewed physicians. The evaluation is mainly relevant for the further development of the tested system while other estimations of the physicians can help to conclude further ethical discussions.

The results are reassuring. The physicians tell us how they used the system, to what extent they trusted the assessment, and what suggestions they have for improvement. Especially for the design of such experiments, these statements are of major importance. For example, it became clear that some of the physicians followed a certain procedure they gave themselves, e.g., by first noting their prediction and then checking the system result ("second opinion") or by first looking at the system result and then challenging it with the available evidence. When

designing further experiments, we can now better decide if we want to impose a certain procedure or not. At the same time, physicians report on the complexity of clinical decision-making in general. They confirm trust to be a prerequisite for being able to make decisions. For an AI-driven support system to be transferred to clinical practice at all, it must therefore be trusted. According to the physicians, the evaluations of the systems must be explainable for this. Not completely, but in a way that the physicians can present the application of the system to the patient, to whom responsibility is always borne.

Without being able to go into more detail about the content of the interviews at this point, we want to emphasize the positive experience that took place with the qualitative survey accompanying the case study. What becomes clear is that through the interviews a connection is made between the profession of the physicians and the researchers. Following the thesis of Noordegraaf [45] that professions have to be more and more in touch with their environment, especially due to technical challenges, we can say from our perspective that qualitative data collection can be a link between physicians and their environment. Our recommendation to future research projects is therefore that they should be accompanied by such or similar studies. These should not only serve as an evaluation of the individual projects but also as part of designing AI decision support systems and as a contribution to the communication process which must be part of the transformations that already take place.

## Conclusion

Targeting computer scientists new to the field of AI in healthcare, this work presented an automatic risk prediction system for severe infections for the next 90 days after kidney transplantation. The performance of this pilot system looked promising in an intrinsic evaluation, although performance was not our focus. Within a small study, we evaluated the influence of our system on the performance of physicians with and without the support of our system. The main findings are that although our risk prediction system on its own outperforms physicians on our small dataset. However, physicians on average do not improve together with the automatic decision support, only junior physicians do. While our system achieves promising results from a technical point of view, our experiments leave us with more open questions than before, such as what a successful decision support system needs to look like to achieve the goal of actually helping physicians treat their patients in a better way. Along with our experiments, we discussed several challenges and lessons learned from a multidisciplinary view. This work highlights that pure performance is not enough to set up a practical tool to support clinicians. Moreover, the presented challenges indicate particularly the need for multidisciplinary work with different stakeholders, as well as shaping the AI-human interaction on different levels.

In future work, we intend to (i) compare the system performance to other systems such as state-of-the-art artificial neural networks, (ii) take the patient's perspective much more into account when designing and evaluating experiments, and (iii) take the next step by integrating the system into the clinical daily business in a prospective manner.

## Author Contributions

**Conceptualization:** Roland Roller, Aljoscha Burchardt, David Samhammer, Sebastian Möller, Bilgin Osmanodja.

**Formal analysis:** Roland Roller, David Samhammer, Simon Ronicke, Sven Schmeier, Bilgin Osmanodja.

**Funding acquisition:** Roland Roller, Aljoscha Burchardt, Sebastian Möller, Peter Dabrock, Klemens Budde.

**Investigation:** Sebastian Möller, Manuel Mayrdorfer.

**Methodology:** Wiebke Duettmann, Manuel Mayrdorfer.

**Project administration:** Sebastian Möller.

**Software:** Roland Roller.

**Supervision:** Aljoscha Burchardt, Sebastian Möller, Peter Dabrock, Klemens Budde, Manuel Mayrdorfer.

**Validation:** Roland Roller, Wiebke Duettmann, Manuel Mayrdorfer, Bilgin Osmanodja.

**Writing – original draft:** Roland Roller, Aljoscha Burchardt, David Samhammer, Simon Ronicke, Sven Schmeier, Sebastian Möller, Peter Dabrock, Klemens Budde, Bilgin Osmanodja.

**Writing – review & editing:** Wiebke Duettmann, Sebastian Möller, Manuel Mayrdorfer.

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
