## [Decision Letter · Decision Letter 0]

17 Oct 2022

PONE-D-22-25782When Performance is not Enough - A Multidisciplinary View on Clinical Decision SupportPLOS ONE

Dear Dr. Roller,

Thank you for submitting your manuscript to PLOS ONE. After careful consideration, we feel that it has merit but does not fully meet PLOS ONE’s publication criteria as it currently stands. Therefore, we invite you to submit a revised version of the manuscript that addresses the points raised during the review process.

We look forward to receiving your revised manuscript.

Kind regards,

Muhammad Fazal Ijaz

Academic Editor

PLOS ONE

Journal Requirements:

2. For studies reporting research involving human participants, PLOS ONE requires authors to confirm that this specific study was reviewed and approved by an institutional review board (ethics committee) before the study began. Please provide the specific name of the ethics committee/IRB that approved your study, or explain why you did not seek approval in this case.

Once you have amended this/these statement(s) in the Methods section of the manuscript, please add the same text to the “Ethics Statement” field of the submission form (via “Edit Submission”)

3. Please ensure that you have specified (1) whether consent was informed and (2) what type you obtained (for instance, written or verbal, and if verbal, how it was documented and witnessed). If your study included minors, state whether you obtained consent from parents or guardians. If the need for consent was waived by the ethics committee, please include this information.

"NO"

"This research was supported by the German Federal Ministry of Education and Research (BMBF) through the project vALID (01GP1903A)."

"The project, and all authors received funding by the German Federal Ministry of Education and Research (BMBF) through the project vALID (01GP1903A).

https://www.gesundheitsforschung-bmbf.de/de/valid-klinische-entscheidungsfindung-durch-kunstliche-intelligenz-ethische-rechtliche-und-10430.php

And no, the sponsor did not play any role in the study design."

6. We note that you have indicated that data from this study are available upon request. PLOS only allows data to be available upon request if there are legal or ethical restrictions on sharing data publicly. For more information on unacceptable data access restrictions, please see http://journals.plos.org/plosone/s/data-availability#loc-unacceptable-data-access-restrictions. 

7. PLOS requires an ORCID iD for the corresponding author in Editorial Manager on papers submitted after December 6th, 2016. Please ensure that you have an ORCID iD and that it is validated in Editorial Manager. To do this, go to ‘Update my Information’ (in the upper left-hand corner of the main menu), and click on the Fetch/Validate link next to the ORCID field. This will take you to the ORCID site and allow you to create a new iD or authenticate a pre-existing iD in Editorial Manager. Please see the following video for instructions on linking an ORCID iD to your Editorial Manager account: https://www.youtube.com/watch?v=_xcclfuvtxQ

Reviewers' comments:

Reviewer's Responses to Questions

**Comments to the Author**

1. Is the manuscript technically sound, and do the data support the conclusions?

Reviewer #1: Partly

Reviewer #2: No

Reviewer #3: Yes

2. Has the statistical analysis been performed appropriately and rigorously? 

Reviewer #1: N/A

Reviewer #2: No

Reviewer #3: I Don't Know

3. Have the authors made all data underlying the findings in their manuscript fully available?

Reviewer #1: Yes

Reviewer #2: No

Reviewer #3: Yes

4. Is the manuscript presented in an intelligible fashion and written in standard English?

Reviewer #1: No

Reviewer #2: No

Reviewer #3: No

5. Review Comments to the Author

Reviewer #1: -Related work needs to be updated more followed by adding references.

-It is recommended to add a table of past work on clinical decision support.

-Improve the quality of figure1.

-Illustrate your clinical decision support model as figure 2.

-Brief on field of nephrology.

-Justify why authors chose this area of Nephrology while many areas also require urgent clinical decision support.

-The statement in the paper required refrences to quote "we present here was developed in a project by some of the

authors of this paper, namely the computer scientists and the clinical experts"

-Contrast your work with the existing or recent works furnished and prove your model to be most significant.

-Showcase your Risk prediction model as data flow diagram.

-Mention the ability of your model to target other areas in the scope for future work.

Reviewer #2: 1. Abstract is too short

2. Introduction need to be improved more add some lines about the motivation about the field of study

3. add major contribution points at the end of the introduction

4. Add literature study section after introduction and cite at least 10 papers also add one table which need to have limitation of the existing field of study.

5. A case study need to be explained more

6. Explain more figure 1

7. Table 1 need to be explained more in detail

8. Add methodology section also work on some more machine learning classifiers

9. Related work is too short

10. Add future directions separate section

11. Add some proper experimental results

12. Overall paper English is too much poor try to improve the language

Reviewer #3: In this paper, authors presented a multidisciplinary view on machine learning in medical decision support systems. However, there are some limitations that must be addressed as follows.

1. This work is not presented correctly. The standard format of the research article should be used: abstract, introduction, related work, methodology, results, and conclusion.

2. The abstract is very short and not attractive. The novelty should be clearly discussed. In addition, In the last lines of abstract, the authors should discuss results and highlight in what %age and in what parameters the proposed methodology is better as compared to existing techniques and what is the overall analysis of proposed methodology.

3. In Introduction section, it is difficult to understand the novelty of the presented research work. This section should be modified carefully. In addition, the main contribution should be presented in the form of bullets.

4. More related work should be included about clinical decision support system (‘An intelligent healthcare monitoring framework using wearable sensors and social networking data’, ‘Automatic detection of Alzheimer’s disease progression: An efficient information fusion approach with heterogeneous ensemble classifiers’, ‘ANAF-IoMT: A Novel Architectural Framework for IoMT-Enabled Smart Healthcare System by Enhancing Security Based on RECC-VC’,’ Fine-Tuned DenseNet-169 for Breast Cancer Metastasis Prediction Using Fast AI and 1-Cycle Policy’). In addition, In the last lines of Literature review, highlight what overall technical gaps are observed in existing works that led to the design of proposed methodology.

5. Captions of the Figures and tables not self-explanatory. These captions should be self-explanatory, and clearly explaining the figure and table. Extend the description of the mentioned figures and tables to make them self-explanatory.

6. Figure 1 is blurred, its quality should be improved.

7. Data analysis section should be extended by including more details.

8. The conclusion section should be revised. In addition, the future work should be included.

6. PLOS authors have the option to publish the peer review history of their article (what does this mean?). If published, this will include your full peer review and any attached files.

Reviewer #1: **Yes: **Dr.Jana Shafi

Reviewer #2: No

Reviewer #3: No

---

## [Author Response · Author response to Decision Letter 0]

15 Dec 2022

Dear reviewers, 

Thank you for your valuable feedback! In the document "response to reviewers (rebuttal letter)", you can find a point by point response to your comments and suggestions.

All the best,

Roland Roller

---

## [Decision Letter · Decision Letter 1]

20 Feb 2023

When Performance is not Enough - A Multidisciplinary View on Clinical Decision Support

PONE-D-22-25782R1

Dear Dr. Roller,

We’re pleased to inform you that your manuscript has been judged scientifically suitable for publication and will be formally accepted for publication once it meets all outstanding technical requirements.

Kind regards,

Muhammad Fazal Ijaz

Academic Editor

PLOS ONE

Additional Editor Comments (optional):

Reviewers' comments:

Reviewer's Responses to Questions

**Comments to the Author**

1. If the authors have adequately addressed your comments raised in a previous round of review and you feel that this manuscript is now acceptable for publication, you may indicate that here to bypass the “Comments to the Author” section, enter your conflict of interest statement in the “Confidential to Editor” section, and submit your "Accept" recommendation.

Reviewer #3: (No Response)

Reviewer #4: All comments have been addressed

2. Is the manuscript technically sound, and do the data support the conclusions?

Reviewer #3: (No Response)

Reviewer #4: Yes

3. Has the statistical analysis been performed appropriately and rigorously? 

Reviewer #3: (No Response)

Reviewer #4: Yes

4. Have the authors made all data underlying the findings in their manuscript fully available?

Reviewer #3: (No Response)

Reviewer #4: Yes

5. Is the manuscript presented in an intelligible fashion and written in standard English?

Reviewer #3: (No Response)

Reviewer #4: Yes

6. Review Comments to the Author

Reviewer #3: The authors have addressed my all comments. I have no further comments. Therefore, this paper can be accepted in its present form.

Reviewer #4: Authors have addressed the recommendations of the reviewers in a reasonable manner. Manuscript in the current form may be processed to next phase of the editorial process.

7. PLOS authors have the option to publish the peer review history of their article (what does this mean?). If published, this will include your full peer review and any attached files.

Reviewer #3: No

Reviewer #4: No

---

## [Editor Report · Acceptance letter]

13 Apr 2023

PONE-D-22-25782R1 

When Performance is not Enough - A Multidisciplinary View on Clinical Decision Support 

Dear Dr. Roller:

I'm pleased to inform you that your manuscript has been deemed suitable for publication in PLOS ONE. Congratulations! Your manuscript is now with our production department. 

Kind regards, 

on behalf of

Dr. Muhammad Fazal Ijaz 

Academic Editor

PLOS ONE